# Style-Aware Radiology Report Generation
# with RadGraph and Few-Shot Prompting

**Benjamin Yan[1], Ruochen Liu[1], David E. Kuo[1], Subathra Adithan[2], Eduardo Pontes Reis[1],**
**Stephen Kwak[3], Vasantha Kumar Venugopal[4], Chloe P. O'Connell[1], Agustina Saenz[5],**
**Pranav Rajpurkar[5,†], Michael Moor[1,†]**

[1]Stanford University, [2]Jawaharlal Institute of Postgraduate Medical Education and Research
[3]John Hopkins University, [4]Aligarh Muslim University, [5]Harvard Medical School,
[†]equal senior authorship

## Abstract

Automatically generated reports from medical images promise to improve the workflow of radiologists. Existing methods consider an image-to-report modeling task by directly generating a fully-fledged report from an image. However, this conflates the content of the report (e.g., findings and their attributes) with its style (e.g., format and choice of words), which can lead to clinically inaccurate reports. To address this, we propose a two-step approach for radiology report generation. First, we extract the content from an image; then, we verbalize the extracted content into a report that matches the style of a specific radiologist. For this, we leverage RadGraph—a graph representation of reports—together with large language models (LLMs). In our quantitative evaluations, we find that our approach leads to beneficial performance. Our human evaluation with clinical raters highlights that the AI-generated reports are indistinguishably tailored to the style of individual radiologist despite leveraging only a few examples as context.

## 1 Introduction

Generating radiology reports from medical images is a crucial task in the field of medical imaging. For human interpreters to write such reports is not only time-consuming and labor-intensive but also requires a high level of expertise (Hartung et al., 2020). Furthermore, these reports are often subject to inter-observer variability, potentially compromising the consistency and accuracy of the reports' findings. As a result, there is a growing interest in methods for automated radiology report generation, which can alleviate these issues and improve overall efficiency of the diagnostic process.

Recent advances in large language models (LLMs) have shown great potential for generating high-quality text, with the ability to customize outputs based on user-specified instructions (Brown et al., 2020b). These models have been utilized to rephrase existing text, paving the way for new approaches to automated report generation in radiology. However, despite the promise of LLMs, their application to this task is not without challenges. One principal concern is their tendency to 'hallucinate', i.e., to generate false information (even if plausible sounding), which can be particularly problematic in high-stakes settings such as when generating reports from medical images.

Previous attempts at automated report generation have largely focused on approaches that aim to produce fully-fledged reports directly from medical images (i.e., a image-to-report modelling task) (Allaouzi et al., 2018; Chen et al., 2020; Wang et al., 2022a). However, this conflates the *content* of the report (i.e., the radiology entities and attributes that are described) with its *style* (i.e., the complement of radiology entities and attributes, or everything needed in terms of language, grammar, and structure to formulate a fully-fledged report, on top of the *content* representation), which limits the flexibility and applicability of such models.

This problem is also reflected in the employed evaluation metrics of existing approaches. They frequently optimize for traditional natural language processing (NLP) metrics, such as BLEU, ROUGE, or METEOR, which measure the similarity of the generated report to a reference report based on lexical overlap. These metrics, while generally useful, may not correlate with the clinical correctness or usefulness of the report (Yu et al., 2022). Previous work has demonstrated that optimizing for clinical metrics that measure the content's relevance, such as the RadGraph scores (Jain et al., 2021), is crucial in generating reports that accurately represent the image findings. The RadGraph score gauges similarity in extracted radiology entities and relations with a ground truth annotation. This emphasizes the report's clinical content as opposed to style or lexical overlap, and makes it an amenable measure of the report's usefulness and completeness in

clinical practice.

Here, we propose to generate radiology reports as a two-step procedure for disentangling report content and style. In the first step, a dedicated model is trained to extract pure content from the image. Specifically, it generates a structured representation (called RadGraph) of the entities and attributes that are present in the image. In the second step, a frozen LLM generates a stylized report from this structured representation. Given a few report examples as context, the LLM can on-the-fly adapt the report style closely to the style of a target radiologist or hospital template for whom the report should be drafted. This model stylization could offer several advantages for radiology workflows: *flexibility*, in generating reports targeted to their readership, such as ones with less jargon that are more accessible to patients; *consistency*, in ensuring clear communication between a radiologist and the referring physician, who may be accustomed to a particular style of reporting; and *emphasis on preferred information*, in highlighting findings most relevant to a specialist's scope of practice (e.g., follow-up on a patch of pneumonia, or the correct location of pacing lead in the heart).

In the first step of our approach, we alter the supervision signal of an image-text model to a serialization of the clinical entities (as captured in RadGraph) and their attributes, rather than the full report text. This step ensures the content extraction model focuses only on generating the report's clinical content, measured by RadGraph score, rather than optimizing for traditional NLP metrics such as BLEU that may not correlate with the report's clinical relevance. By generating the clinical content first, we prioritize the report's clinical usefulness over its stylistic quality, which can be improved and even personalized in the second step.

For the second step, we leverage GPT-3.5 (a generative LLM developed by OpenAI) to transform the predicted serialization of clinical entities and attributes from the image into a styled radiology report (Brown et al., 2020a). These models have shown great promise in generating high-quality text (Anil et al., 2023), including summaries and paraphrases of existing text (Li et al., 2023a), which we can use to inject a hospital-specific style, as well as to enhance readability and understandability. By separating the content generation and style injection steps, we can ensure the model optimizes for the relevant criteria for each step, resulting in a

high-quality report.

Our approach addresses the limitations of end-to-end image-to-report generation, as the LLM does not need to suggest facts about the image. Instead, it can focus on rephrasing the structured set of entities and attributes that were first derived from the image and infuse the desired style into the report. At prediction time, the LLM in-context learns a serialization-to-report mapping from a few examples of a target report writing style. Our method not only offers a novel solution to the challenges posed by previous approaches but also enhances the customization and adaptability of radiology reports generated by LLMs.

This paper makes the following contributions:

- First, we develop a method for extracting structured content—that is, a serialized version of RadGraph—from radiology images.

- Second, we propose a strategy for generating stylized reports from this extracted content by means of in-context learning from few style-specific example reports.

- Third, our overall system (combining content extraction and style generation) achieves competitive performance at radiology report generation.

- Fourth, in a human style evaluation clinical experts were not able to distinguish real reports from AI-generated ones that were adapted to the writing style of individual radiologists.

## 2 Related Works

### 2.1 Medical Report Generation

Medical report generation (MRG) has seen a recent insurgence in the field of medical AI. Early works (Allaouzi et al., 2018) adhere to the methods of image captioning models (Vinyals et al., 2015; Xu et al., 2015), leveraging deep CNNs to extract image features and RNNs to generate text descriptions in an encoder-decoder fashion. Meanwhile, emerging in several works was an auxiliary classification task to predict certain medical abnormalities, with the aim of more structured guidance for report generation (Shin et al., 2016; Wang et al., 2018; Yin et al., 2019; Yuan et al., 2019). Later, the use of the attention mechanism in MRG systems became increasingly prevalent (Jing et al., 2017; Chen et al., 2020). To further bridge visual and linguistic modalities while incorporating med-

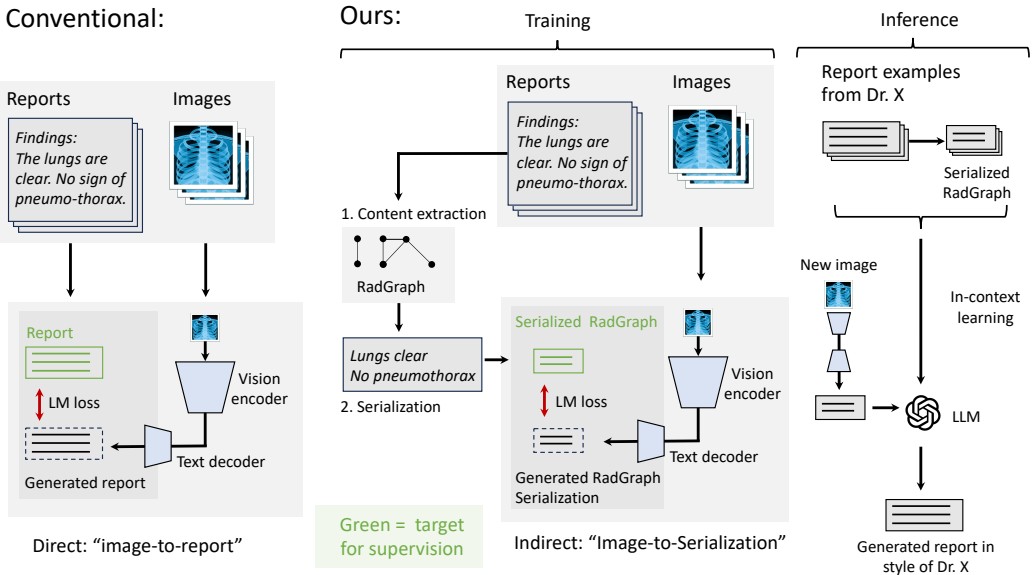

Figure 1: Overview of our two-step pipeline for generating radiology reports (right panel) as contrasted with the standard "image-to-report" strategy for directly generating radiology reports from X-ray images (left panel). During training, we consider a set of paired chest X-ray images and radiology reports. We preprocess the reports by extracting its content in the form a graph representation (RadGraph). We serialize the RadGraph into a condensed summary of the clinical content of the report. From there, this RadGraph serialization is used as the supervising signal in an image captioning model. For illustration purposes, we omit to display the text encoder that augments both captioning models with clinical context. At inference time, a new chest X-ray is fed through the model to generate a serialized RadGraph. By providing few example reports of a specific radiologist (and extracting their RadGraphs), an LLM can in-context learn to verbalize the structured content extracted from the image into a fully-fledged report in the style of the target radiologist.

ical domain knowledge, various kinds of knowledge graphs have been explored for use (Li et al., 2019, 2023b; Zhao et al., 2021; Zhang et al., 2020; Liu et al., 2021). In our study, we use the classic encoder-decoder architecture, as it is a common denominator of many MRG approaches.

Subsequent studies acknowledge the constraints of traditional natural language generation metrics when assessing medical reports, prompting a growing emphasis on ensuring clinical accuracy. RadGraph (Jain et al., 2021) is a dataset of entities and relations in full-text chest X-ray radiology reports based on a novel information extraction schema. (Yang et al., 2022) adopts the knowledge graph provided by RadGraph as general knowledge embedding. (Wang et al., 2022b) employed a classification loss for medical concepts provided by RadGraph. (Delbrouck et al., 2022) improves the factual completeness and correctness of generated radiology reports with a well-designed RadGraph reward. Most existing methods develop metrics around RadGraph and integrate that into the objective function; our approach, on the other hand, directly trains the model to generate a serialized (i.e., text) representation of RadGraph as we decouple the content and style generation and focus solely on the clinical correctness during the content generation stage. Additionally, serialized RadGraphs allow us to juxtapose dense content representations with stylized reports which enables style adaptation via in-context learning.

## 2.2 Large Language Models

Recent watersheds such as the Transformer architecture (Vaswani et al., 2017), generative pre-training objectives (Radford et al., 2018), and increased computing power have facilitated the training of large language models comprising billions of parameters (Brown et al., 2020a; Chowdhery et al., 2022; Touvron et al., 2023). These advances have significantly burgeoned model capability in tasks such as translation, summarization, and generating long text that closely resembles human language.

In 2021, OpenAI announced GPT-3 (Brown et al., 2020a; Floridi and Chiriatti, 2020), a generative language model featuring an unprecedented 175 billion parameters. Their study introduces in-

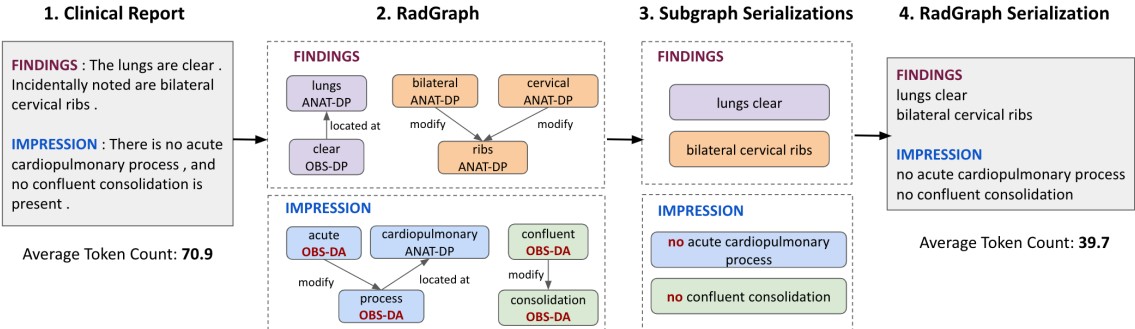

Figure 2: Overview of the RadGraph serialization. In the RadGraph entities, ANAT-DP, OBS-DP, and OBS-DA refer to present anatomical, present observational, and absent observational entities, respectively. The token counts refer to the averages across the test set.

context learning, the ability of LLMs to learn to perform a task simply by being provided a few examples as context—without any parameter updates.

LLMs have also demonstrated great potential in the medical domain. For instance, GPT-4 has been employed to post-hoc transform free-text radiology reports into structured reports (Adams et al., 2023). As for LLM-based MRG systems, Wang et al. (2023) utilized ChatGPT to generate medical reports based on features extracted by neural networks (e.g., disease classifier). However, their approach does not exploit in-context learning and thus has limited control over format and style.

## 3 Methods

### 3.1 Dataset

Our study makes use of the MIMIC-CXR (Johnson et al., 2019) and RadGraph (Jain et al., 2021) datasets. MIMIC-CXR is a large dataset of 377 110 chest radiographs imaged at Beth Israel Deaconess Medical Center from 227 835 studies, with free-text medical reports. The RadGraph dataset is publicly available and comprises radiology text reports and corresponding knowledge graphs (Figure 2).

To preprocess reports, we extract only *Findings* and *Impressions*, as other sections contain details that cannot be readily referenced from the image, such as patient demographics or lab results from different procedures. *Findings* refer to the direct observations from the image (e.g., opacity of lungs, catheter placement), while *Impression* summarizes the most urgent inferences and diagnostically relevant findings (e.g., presence of pneumonia).

### 3.2 RadGraph

As notation, a *RadGraph* refers to any knowledge graph within the titular dataset. The nodes of a

RadGraph are either anatomical (e.g, lungs, cardiomediastinal, carina) or observational entities (e.g., acute, abnormality). The edges are directed and heterogeneous, capturing three types of relations—modify, located at, suggestive of—between entities. Nodes and edges are automatically obtained via a named entity recognition and relation extraction model on MIMIC-CXR reports, employing the DYGIE++ framework from Wadden et al. (2019). This embodies the *Report → RadGraph* preprocessing step, or *content extraction* in (Figure 1). We distinguish this from *content generation* (Section 3.4), which is the transformer-based prediction of serialized content based on the image.

### 3.3 RadGraph Serialization

We serialize each RadGraph into a structured text representation (Figure 2), which serves as the supervision label for the content generation model (Section 3.4). This serialization acts as a dense content representation whose advantages over the free-text report include conciseness and the pruning of non-semantic content (e.g., style-filler words, radiologist-specific phrasing) from the report. The aim is to focus the model purely on generating the content backbone at this step, and defer the style injection to a later stage of the pipeline (Section 3.5).

To exploit the graph structure of the input RadGraph, we first extract the weakly connected components (step (3) in Figure 2). These are the maximal subgraphs where any two nodes can be reached through a path of undirected links. Each component is thus a separate network of spatially and medically related entities, segmenting the chest X-ray's content into distinct regions of interest that can be serialized in parallel.

For each component, we create a text span of all

labelled entities. The keywords *no* and *maybe* are prepended to absent and uncertain entities, respectively. The entity ordering follows the syntax of the corresponding tokens in the report to ensure readability and lexical fidelity. When the report contains both *Findings* and *Impression* sections (Figure 2), we analogously stratify the components based on their referenced location in the free-text report. This is crucial as the content representation should carefully distinguish between factual image information (*Findings*) and clinical inferences (*Impressions*), even if strongly supported.

Within *Findings* and *Impression*, the components are concatenated and separated by delimiters. The two sections are combined into a single text, which is the report serialization and densely characterizes the full chest radiograph. In the case where the report is not bipartite (e.g., only *Impressions*), we unify the components under a singular section.

## 3.4 Content Generation Model

For the content generation model, we leverage the encoder-decoder architecture that has been widely employed in image-to-text systems. An image encoder takes chest X-ray images as the input and encodes them into a visual feature representation. In parallel, a text encoder reads clinical documents, such as doctor indications, and transforms the textual content into dense feature vectors.

The visual and text embeddings are then added together and passed through a LayerNorm operation to form contextualized embeddings. The fused embeddings are then fed into the report decoder, which generates serialized RadGraph word-by-word. The main architecture is adapted from (Nguyen et al., 2021), but we simplified it by removing the classifier and the interpretation module to eliminate as many potential confounders as we can since our goal is to evaluate the influence of the supervision signal.

**Image Encoder** We adopt a DenseNet-121 (Huang et al., 2017) model pre-trained on the ImageNet dataset as the image encoder. For each input chest X-ray image $I$, it extracts a feature vector $d \in R^e$ where $e$ is the embedding dimension. If an imaging study consists of more than one image, the feature will be obtained via max-pooling across all feature vectors extracted from each image. $d$ is then transformed into $n$ low-dimensional disease representations $D_{img} \in R^{n \times e}$.

**Text Encoder** We use a Transformer encoder to extract features $H = \{h_1, h_2, ..., h_l\}$ from the clinical document text input with length $l$ consisting of word embeddings $\{w_1, w_2, ..., w_l\}$ where $w_i \in R^e$ is the vector representation of the $i$-th word in the text, $e$ is the embedding dimension and $h_i \in R^e$ is the attended features of the $i$-th word to other words in the input document. The features are then transformed into summarization denoted as $Q = \{q_1, q_2, ..., q_n\}$ representing a set of $n$ disease-related topics (such as pneumonia or atelectasis) to be queried from the document, as proposed in (Nguyen et al., 2021),

$$D_{txt} = \text{Softmax}(QH^T)H \qquad (1)$$

where matrix $Q \in R^{n \times e}$ is constructed by vertically stacking $\{q_1, q_2, ..., q_n\}$ where each vector $q_i$ is initialized with random values and subsequently refined through the attention process, and $H \in R^{l \times e}$ is formed by stacking $\{h_1, h_2, ..., h_l\}$.

**Fused Embedding** We obtain the final, contextualized embedding $D \in R^{n \times e}$ by entangling the visual embedding $D_{img}$ and text embedding $D_{txt}$,

$$D = \text{LayerNorm}(D_{img} + D_{txt}) \qquad (2)$$

where $D$ will be the input for the report decoder.

**Report Decoder** We use the Transformer as the backbone of our report decoder to generate long, robust text. A total of 12 decoder components are stacked together where each component consists of a masked multi-head self-attention component followed by a feed-forward layer. The report generation objective (Equation 3) is defined as the cross-entropy loss between predicted words $p$ and ground truth $y$. Here, we denote $p_{ij}$ as the confidence of selecting the $j$-th word of vocabulary $V$ in the $i$-th position in the generated text, and $y_{ij}$ as a binary indicator of whether the $j$-th word appears in the $i$-th position of the ground truth.

$$L = -\frac{1}{l} \sum_{i=1}^{l} \sum_{j=1}^{v} y_{ij} \log(p_{ij}) \qquad (3)$$

## 3.5 Style Generation Step

We describe the process of prompting a pre-trained LLM to generate reports from the serialization. This enables adapting the generation to a specific style by supplying the LLM with relevant

in-context examples. Each example is a pair: serialization $s_i$ from the RadGraph, and corresponding ground truth report $r_i$ under the desired style.

We use the *gpt-3.5-turbo* model from OpenAI, a dialogue-based LLM that accepts a sequence of messages as input, rather than a singular text prompt. This is useful for inserting style pairs seamlessly; we relay them as a back-and-forth conversation between *user* and *assistant* roles, where a *user* role supplies serialization $s_i$ and an *assistant* responds with target report $r_i$ for the language model to learn in-context.

The chain of $K$ examples $\{(s_i, r_i)\}_{i=1}^K$ is prefaced by a system role message indicating the LLM should act as the report-generating *assistant*, establishing its specific task within the radiology-based dialogue. The remaining prompt is structured as $s_1 : r_1, s_2 : r_2, \cdots, s_K : r_K, \hat{s}_{\text{eval}} :$. At the end, the model is given an evaluation serialization $\hat{s}_{\text{eval}}$ predicted from the chest X-ray image using our content generation model, cuing the LLM to generate the corresponding report prediction $\hat{r}_{\text{eval}}$ next. Note that in the zero-shot case, the prompt is just $\hat{s}_{\text{eval}} :$, with no preceding context examples.

### 3.6 Evaluation Metrics

We present a comprehensive quantitative evaluation of our approach with commonly-used metrics concerning both language fluency and clinical accuracy. For each metric, we display the mean $\overline{x}$ across the $n$ test reports, as well as the $95\%$ confidence interval ($\overline{x} \pm 1.96 \cdot \frac{\sigma}{\sqrt{n}}$).

**Natural Language Generation Metrics (NLG)** As for classical NLG metrics, we compute BLEU-2 and BERT scores. However, these metrics have relevant limitations due to focusing on lexical similarity or general (non-clinical) semantics, respectively, thereby lacking in the assessment of clinical similarity (for more details, see Section A.5).

**Clinical Accuracy Metrics** CheXbert vector similarity extends beyond BERT by utilizing CheXbert, a model trained specifically on datasets comprising chest X-rays. It computes the cosine similarity between the indicator vectors of 14 pathologies that the CheXbert labeler extracts from machine-generated and human-generated radiology reports. It is designed to evaluate radiology-specific information but its evaluation is limited to 14 pathologies. To address this limitation, we also adopt RadGraph F1 (Yu et al., 2022) that calcu-lates the overlap in clinical entities and relations extracted by RadGraph from both machine-generated and human-generated reports. In this study, we lay emphasis on clinical metrics because we aim to generate reports in different styles while keeping accurate clinical information instead of reports that lexically match the ground truth in the dataset.

### 3.7 Experimental Setup

We conduct four experiments to scrutinize each individual step and overall performance of our proposed strategy for report generation. This includes:

1. *Image to Serialization:* We evaluate content generation model in terms of comparing the content of the generated serialization with the ground truth report.

2. *Serialization to Report:* Conditioning on a strong image-to-serialization model, we evaluate LLM-based style injection by using the ground-truth RadGraphs as input to the LLM.

3. *End-to-end report generation:* We evaluate the pipeline end-to-end, feeding the image and clinical context through the content generation model and the generated serialization through the LLM.

4. *Human style evaluation:* To evaluate the style quality, we let physicians rate sets of 4 radiology reports where 3 were written by the same radiologists and 1 was AI-generated by our method following the style of the radiologist. The goal for the physicians is to detect the AI-generated report and to justify their choice.

**Baseline** The baseline model is adapted from (Nguyen et al., 2021), sharing the same architecture as our image-to-serialization model (Section 3.4), but its supervision target is the full report (*Findings* and *Impressions*) instead of serialization. The training involves an identical parameter set as above.

Additional experimental details (training, hyper-parameter search, and infrastructure) are provided in Section A.1.

## 4 Results

### 4.1 Image to Serialization

We train two content generation models, one with the full report as the supervision target, and the other with the serialized RadGraph. In table 2, we present the RadGraph F1 evaluation result on the

| Method | Examples | RadCliQ ($\downarrow$) | RadGraph F1 ($\uparrow$) | CheXbert ($\uparrow$) | BLEU ($\uparrow$) | BERT Score ($\uparrow$) |
|--------|----------|---------|--------------|----------|------|------------|
| Baseline | — | $3.553 \pm 0.032$ | $0.186 \pm 0.005$ | $0.352 \pm 0.008$ | $\mathbf{0.184} \pm 0.005$ | $\mathbf{0.378} \pm 0.006$ |
| Ours | 0 | $3.527 \pm 0.027$ | $\mathbf{0.228} \pm 0.004$ | $\mathbf{0.394} \pm 0.008$ | $0.162 \pm 0.003$ | $0.333 \pm 0.004$ |
|  | 1 | $3.512 \pm 0.028$ | $0.226 \pm 0.005$ | $0.393 \pm 0.008$ | $0.170 \pm 0.003$ | $0.343 \pm 0.004$ |
|  | 5 | $3.489 \pm 0.028$ | $0.224 \pm 0.004$ | $0.391 \pm 0.008$ | $0.177 \pm 0.003$ | $0.354 \pm 0.004$ |
|  | 10 | $\mathbf{3.485} \pm 0.028$ | $0.224 \pm 0.005$ | $0.390 \pm 0.008$ | $0.180 \pm 0.003$ | $0.357 \pm 0.004$ |

Table 1: Results for the end-to-end report generation. The baseline is the direct image-to-report transformer, i.e., the content generation model trained to predict reports directly from image. Note that RadCliQ (lower is better) is the most important metric as it best corresponds with radiologists' quality assessment (Yu et al., 2022). Best row is in bold. Rows leveraging in-context examples are grayed out, whereas the first two rows use the exact same input data.

MIMIC-CXR test set, comparing the outputs from both models against the ground truth full report.

The comparison is suitable as RadGraph F1 is agnostic of general lexical similarity, measuring only overlap in radiology entities and relations extracted from the text. We find the model trained on the serialized RadGraph outperforms the model trained on the full report. This verifies our assumption that switching the supervision signal to the serialization would help focus the model on generating clinical content with greater accuracy and saliency.

| Supervision Target | RadGraph F1 |
|--------------------|-------------|
| Full Report | $0.186 \pm 0.005$ |
| Serialized RadGraph | $0.221 \pm 0.004$ |

Table 2: RadGraph entity and relation F1 score of models trained on full report versus serialized RadGraph.

### 4.2 Serialization to Report

For selecting examples, we draw randomly from the train split. This avoids patient overlap that could unfairly benefit performance on test cases. We compare the LLM-generated reports against the ground truth reports. Results are provided (Table 3) for varying numbers of in-context examples. We observe strong performance across all metrics, including a 0.722 RadGraph F1 mean in the zero-shot regime. Furthermore, lexical overlap metrics such as BLEU and BERT score saw noticeable improvement with more examples (20.2% and 15.7% increases, respectively, from 0 to 10 examples). This aligns with the aim of in-context learning to improve the style fidelity of generated reports.

### 4.3 End-to-End Report Generation

We evaluate the end-to-end performance from chest X-ray to report by concatenating the content generation step with the style generation step. The results are presented in Table 1. We observe that our two-step model surpasses the baseline (direct

image-to-report model) in clinical accuracy metrics (CheXbert similarity, RadGraph F1, RadCliQ), even in the zero-shot style generation case. This illustrates greater accuracy in extracting radiology content from the chest X-ray, the central focus of separating *content generation* from *style injection*. Notably, RadCliQ is a composite metric found by Yu et al. (2022) to best correlate with quality judgement of human radiologists.

The primary contribution of more examples is improving lexical and non-clinical NLG metrics (BLEU, BERT Score), similar to our findings in the *Serialization → Report* step. However, they remain slightly lower than those of the baseline. A potential explanation is that during training, the baseline is directly supervised with full reports, while only the extracted content is available to our content generation model—with the report itself synthesized with an external, pre-trained LLM.

### 4.4 Human Style Evaluation

Four board-certified radiologists were instructed to write chest X-ray reports in their usual style from 40 randomly selected chest X-ray images from the MIMIC-CXR test set (Figure 4). As several chest X-rays in the MIMIC-CXR dataset were within normal limits or interval follow ups on intensive care patients, duplicate or near-duplicate chest X-ray reports were removed upon manual inspection.

The style generation step described in Section 3.5 was then used to produce AI-generated chest X-ray reports in the style of each of the four radiologists. From these radiologist-generated and AI-generated chest X-ray reports, we created 23 sets of four chest X-ray reports for style evaluation by physician evaluators (i.e., the target audience for written radiology reports). In each set, three reports (corresponding to three different chest X-rays) from the same radiologist, and one AI-generated report (from a fourth chest X-ray) in the style of that radiologist were presented in random

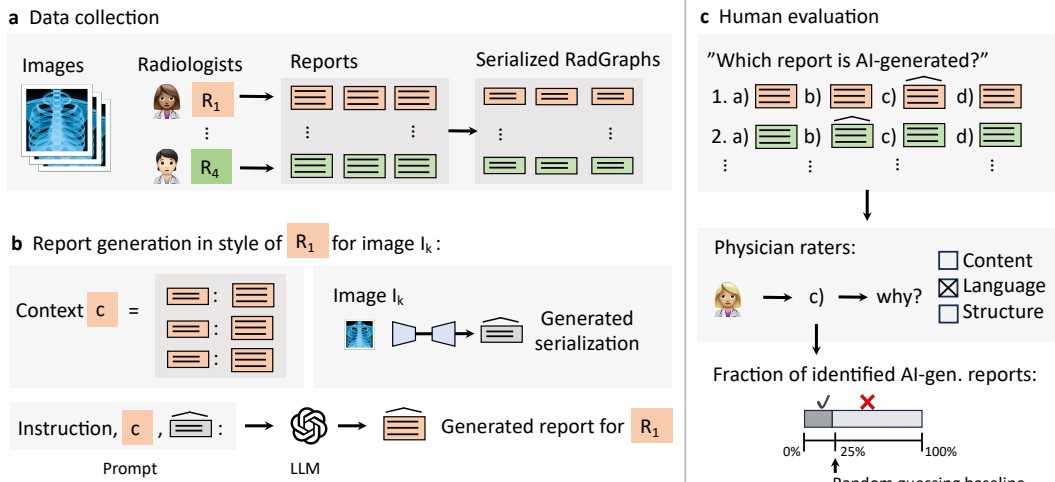

**a** Data collection

Images  Radiologists  Reports  Serialized RadGraphs

**b** Report generation in style of $R_1$ for image $I_k$:

Context $c$ =

Image $I_k$ → Generated serialization

Instruction, $c$ , : → LLM → Generated report for $R_1$

Prompt  LLM

**c** Human evaluation

"Which report is AI-generated?"

1. a) b) c) d)
2. a) b) c) d)

Physician raters:

→ c) → why? → ☐ Content ☒ Language ☐ Structure

Fraction of identified AI-gen. reports:

0%  25%  100%

Random guessing baseline

Figure 3: Overview of human expert evaluation of the style generation. In panel **a**, a set of radiologist write radiology reports in their own style for $n$ chest X-ray studies. For each written report, we extract the RadGraph representation and serialize it into the text representation. Panel **b** illustrates how we AI-generate a radiology report in the style of a specific radiologist, $R_1$, for an input image $I_k$. To in-context learn the style of $R_1$, for $k$ shots we provide a context of $k$ input / output pairs of serialized RadGraphs (dense content representation) together with the corresponding stylized reports as written by $R_1$. Next, to generate a report for a given image $I_k$, we feed the image through our image-to-serialization model and use the generated RadGraph serialization (as indicated with a hat) as the final query of the prompt. The LLM is then prompted to return a generated report in the style of $R_1$. Panel **c** illustrates the human evaluation. Physician raters are provided with problems each consisting of 4 reports, 3 written by one radiologist and 1 AI-generated in the same radiologist's style.

order. Three physician evaluators were asked to identify the AI-generated report out of the four reports and indicate whether report content, language, or report structure contributed to their choice (for more details, refer to Section A.3).

A one-sided Z-test was performed for the proportion of AI-generated reports correctly identified by the three evaluators with a null hypothesis of 25%, corresponding to random chance, and an alternative hypothesis of >25%, corresponding to AI-generated reports being identified at a rate greater than that of random chance. Evaluators A, B, C correctly identified 5 out of 23 (21.7%), 5 out of 23 (21.7%), and 4 out of 23 (17.4%) AI-generated chest X-ray reports, respectively, for a mean accuracy of 20.3%. The one-sided Z-test produced p-values of 0.648, 0.648, and 0.832 for Evaluators A, B, and C respectively, and 0.835 when pooling all evaluators together.

One potential explanation for why evaluators identified <25% of AI-generated reports (worse than random), is variance in radiologist style. For example, our radiologists would sometimes alternate between using parentheses to highlight the chest X-ray view (e.g., "Chest X-ray (PA and lateral views):") and not using any parentheses (e.g.,

"Chest X-ray PA and lateral views:"). Human evaluators may use particular heuristics to call a report AI-generated, when faced with similar appearing reports. However, if they are not reflective of the truth, this may result in accuracy less than random.

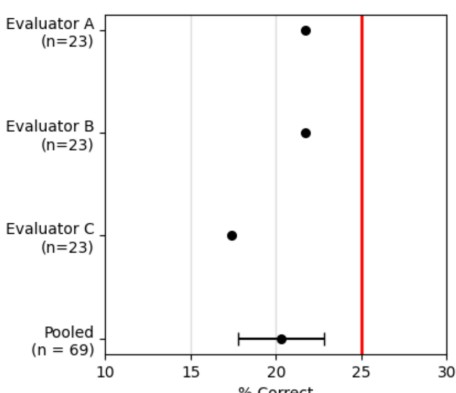

Figure 4: Human Style Evaluation: All evaluators identified the AI-generated chest X-ray report at rates lower than 25% (corresponding to random chance).

## 5 Discussion

We presented a novel approach for radiology report generation that disentangles the report's content from its style. Our experiments showed that our

method offers several advantages over the prevailing paradigm of direct image-to-report modeling. First, training a content extraction model to predict a serialized RadGraph represention from the input image helps focusing the model on the clinically relevant content which is reflected in improved performance metrics (Table 2). Second, when concatenating the content generation step with the style injection step, we observe favourable performance compared to the direct image-to-report baseline (Table 1). Third, by in-context learning the radiologist-specific mapping from serialized RadGraph to report, our method enables the generation of high quality reports that are tailored to the individual radiologist to the degree of indiscernability using just a few example reports. To ultimately determine the clinical utility of our method, deployment studies will be an exciting venue for future work. Another promising direction would be to expand our two-step report generation paradigm to other modalities such as radiology mammograms and magnetic resonance imaging (MRI).

## 6 Limitations

A limitation of our approach is that it relies on the accuracy and effectiveness of the initially extracted RadGraph representation, as this serves as a key supervision signal in our model. Notably, RadGraphs are extracted at inference time using an automated model rather than manual expert labelling. The model achieves high performance in entity and relation extraction (Jain et al., 2021) but is susceptible to error, particularly with report inputs that contain rarer medical entities or ambiguous observations.

Furthermore, due to our employed LLM being served by a third party (OpenAI), reproducing our results comes at the financial costs of using Azure OpenAI's service. Another consequence of relying on this service is that we cannot guarantee the deterministically exact reproduction of our findings, as the served LLM models may change and potentially degrade over time—for instance, if models are replaced by distilled versions thereof.

## 7 Ethics Considerations

A principal ethical consideration is the de-identified, credentialed medical data we worked with. In particular, responsible usage policy of the MIMIC-CXR dataset (Johnson et al., 2019) prohibits sharing access to third parties. This disqualifies the use of ChatGPT or large language model

APIs for prompting models to generate radiology reports from our content representations. However, cloud-based services are allowed, including the Azure OpenAI platform, which we use for deploying and prompting the *gpt-3.5-turbo* model. A stipulation is the monetary service costs, which are counted at a rate per one thousand tokens. These can pose financial barriers to equitable access to high-end language models (which are typically more expensive), as well as usage at scale.

Furthermore, as discussed in the Introduction section, the use of large language models is accompanied by their risks of "hallucination" and generating false or misleading content. These risks can be amplified in the critical setting of medical report generation. To mitigate them, we prompt the LLM not to synthesize medical content, but rather to rephrase existing content into readable, stylized prose. We assist the model through providing content serialization to report pairs as in-context examples. This is our *style injection* step, which we intentfully separate from the *content generation* step to reduce the opportunity for LLM hallucination when generating the full report prediction.

## 8 Acknowledgements

This project was supported by AWS promotional credits.

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

## A  Appendix

### A.1  Further experimental details

In the following, we provide additional details about our experimental setup including information about the model training and used infrastructures.

**Training**  The image-to-serialization model is trained and evaluated on the official train/test split of the MIMIC-CXR (v2.0.0) dataset with $213\,501$ and $2\,799$ chest X-ray reports, respectively. The ground-truth RadGraph serialization associated with each study is taken from the RadGraph dataset. The model is trained for 25 epochs, with a batch size of 16, a learning rate of 0.0001, an embedding size of 512, a weight decay of 0.001, a dropout rate of 0.1, and 12 transformer blocks in the decoder.

**Hyperparameter Search**  We optimized our hyperparameters for the content generation model through grid search with the help of WandB. Figure 5 visualizes the performance of the model with a different set of hyperparameters. We search the learning rate from [0.001, 0.0001], embedding dimension from [256, 512], number of transformer decoder components from [6, 12], and weight decay rate from [0.001, 0.0001, 0].

**Infrastructure**  The image-to-serialization model was trained on an AWS g5.2xlarge instance with one NVIDIA A10G Tensor Core GPU. It was trained for 25 epochs or roughly 50 hours. Evaluation was dispatched on two NVIDIA RTX A4000 GPUs with 16 GB of memory. We use the Azure OpenAI service to access the *gpt-3.5-turbo* language model, with a cloud-based deployment.

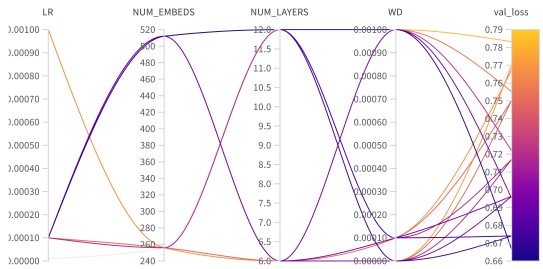

Figure 5: Sweeps of our hyperparameter search: we use validation loss as the metric to search for best hyperparameters.

## A.2 Serialization to Report Results

Table 3 shows the quantitative results of our Serialization to report generation task as described in Section 4.2.

## A.3 Human Style Evaluation

Here, we provide further results on the human style evaluation of the generated reports. To reiterate the approach, three physicians were tasked to identify an AI-generated report within a set of four radiology reports: three were written by the same radiologist and the fourth one (appearing in random order) was generated using our approach using in-context examples from the same radiologist.

Figure 6 shows the cumulative count of explanations stratified by clinical evaluator (rows), as well as by whether the selection was correct or incorrect (columns). Language (e.g., word choice, grammar, and/or writing style) was the primary heuristic that evaluators used to decide whether a report was human-generated or AI-generated, followed by content (e.g., missing important details or including extraneous details), and structure (e.g., different use of numbering or section headings).

## A.4 LLM Prompting

Here, we illustrate a template of our dialogue-based prompt for the LLM to generate a report prediction, with 2-shot learning to adapt to a report style. For notation, <SERIALIZATION $i$> and <REPORT $i$> are placeholders for the $i$-th serialization and corresponding ground truth report, respectively, and <EVAL SERIALIZATION> is the serialization predicted from an image in the evaluation set.

- **System:** You are a helpful assistant that generates chest x-ray reports from key words.

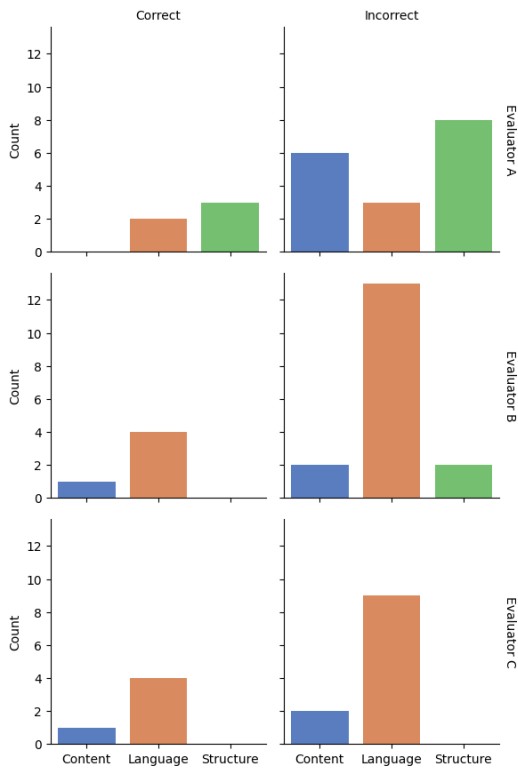

Figure 6: Human Style Evaluation Explanations: When style evaluators attributed their choice of AI-generated report, they attributed it primarily to language and structural differences between human-generated and AI-generated reports.

- **User:** Generate a chest x-ray report from the following key words:
  <SERIALIZATION 1>

- **Assistant:** <REPORT 1>

- **User:** Generate a chest x-ray report from the following key words:
  <SERIALIZATION 2>

- **Assistant:** <REPORT 2>

- **User:** Generate a chest x-ray report from the following key words:
  <EVAL SERIALIZATION>

The LLM will proceed to generate its report prediction from the evaluation serialization, which we compare against the corresponding ground truth report in the test set.

## A.5 Natural Language Generation Metrics

BLEU-2 is widely used in machine translation tasks; it measures the similarity between a candidate translation and one or more reference trans-

| Examples | RadCliQ ($\downarrow$) | RadGraph F1 ($\uparrow$) | CheXbert ($\uparrow$) | BLEU ($\uparrow$) | BERT Score ($\uparrow$) |
|---|---|---|---|---|---|
| 0 | $1.357 \pm 0.030$ | $0.722 \pm 0.006$ | $0.843 \pm 0.007$ | $0.388 \pm 0.005$ | $0.568 \pm 0.005$ |
| 1 | $1.234 \pm 0.031$ | $0.723 \pm 0.007$ | $0.847 \pm 0.007$ | $0.423 \pm 0.005$ | $0.614 \pm 0.005$ |
| 5 | $1.125 \pm 0.030$ | $0.731 \pm 0.006$ | $0.852 \pm 0.007$ | $0.460 \pm 0.006$ | $0.652 \pm 0.005$ |
| 10 | $\mathbf{1.107} \pm 0.030$ | $\mathbf{0.733} \pm 0.006$ | $\mathbf{0.853} \pm 0.007$ | $\mathbf{0.466} \pm 0.006$ | $\mathbf{0.657} \pm 0.005$ |

Table 3: Results for the *Serialization → Report* generation task, using the ground truth serialization in place of the predicted one. The goal of this experiment is to evaluate the style generation in isolation, assuming a very strong content generation step (here the ground truth serialization of RadGraph). Note a lower RadCliQ score is better. The best row is bolded. Leveraging more in-context examples leads to better performance across all metrics.

lations by comparing their bigram overlaps. Although BLEU-2 is a fast and reliable metric, it possesses a few limitations, e.g., it does not take into account synonymous words and the proper use of grammar. We thus also adopt the BERT score, a recently proposed metric for assessing the quality of machine-generated text. It takes into account the semantic similarity between the generated text and reference text by calculating the contextual embeddings of both texts using BERT and measuring their cosine similarity. Nevertheless, BERT is a general-purpose metric and not at all optimized for capturing clinical semantics and radiological findings. This is why the included clinical accuracy metrics are most salient to our quantitative evaluation of radiology report generation.