# OpenReview forum: "Style-Aware Radiology Report Generation with RadGraph and Few-Shot Prompting"
_EMNLP/2023/Conference — EMNLP 2023 Findings_

### Official Review · Reviewer_SCYV · 2023-07-30

**Soundness:** 3

**Excitement:**

4: Strong: This paper deepens the understanding of some phenomenon or lowers the barriers to an existing research direction.

**Paper Topic And Main Contributions:**

The paper proposes a two-step approach for generating radiology reports that match a physician's style from medical images. The authors first extract content from images and then verbalize it into a report matching the specific radiologist's style. They leverage RadGraph, a graph representation of reports, along with large language models (LLMs). The paper reports positive quantitative evaluations and highlights that the AI-generated reports are tailored to individual radiologist styles.

**Questions For The Authors:**

I strongly recommend the authors to address the following concerns.
1. The work argues that existing methods, which generate reports directly from images, may lead to clinically inaccurate reports due to conflating content and style. Could you give some examples or evidence to prove it?
2. Why is it more accurate to write a report that matches a physician's style? What is the advantages of usefulness of matching a physician's style?
3. How to serialize the RadGraph into a condensed summary of the clinical content of the report?
4. How to ensure/define the order of entities in the generated Serialized RadGraph?
5. For example, in Figure 2, how to obtain the "acute cadiopulmonary process"? How to obtain the "no"?
6. How to obtain the Serialized RadGraph from the Graph?
7. What is the purpose of introducing a text encoder? If I understand correctly that the input is just the image, what and where is the input text?
8. What is the full input of GPT-3.5 under both few-shot and zero-shot settings? Could you give an example to illustrate the full input and full output of GPT-3.5?
9. How many examples are used to prompt GPT-3.5?
10. Could you give more details about the Image to Serialization task? What are the input and output of this task? What is the meaning of comparing the content of the generated serialization with the ground-truth report? Why is only the F1 score calculated? What are the results of other metrics, e.g., BELU? Therefore, it is important to clarify what are the input and output of this task.
11. In Table 2, what is the baseline model? Pure Transformer or R2Gen? How many Transformer layers and the number of parameters are used in the baseline?
12. Besides, while the paper reports beneficial performance, a detailed comparison with more existing image-to-report methods is missing. Including such a comparison would provide a clearer understanding of the proposed approach's advantages.
13. When evaluating the AI-generated reports, are the physicians given the test set CXR images?
13. The criteria used in human study is subjective. Authors are encouraged to introduce more information about three clinicians, i.e. working experience and expert title. Senior and junior clinicians may find differences when considering whether the same report is AI-generated or not.
14. I strongly recommend the author to further evaluate the quality of the generated reports, instead of asking the physician to judge whether the report is AI-generated or not.

**Reasons To Accept:**

1. The proposed approach is sound and the utilization of RadGraph is well-motivated. The idea is interesting.

2. The beneficial performance shown in the evaluations validates the effectiveness of the proposed approach.

**Reasons To Reject:**

1. Several claims should be proved.
- The work argues that existing methods, which generate reports directly from images, may lead to clinically inaccurate reports due to conflating content and style. Could you give some examples or evidence to prove it?
- Why is it more accurate to write a report that matches a physician's style? What is the advantages of usefulness of matching a physician's style?

2. More model details should be provided and clarified.
- How to serialize the RadGraph into a condensed summary of the clinical content of the report?
- How to ensure/define the order of entities in the generated Serialized RadGraph?
- For example, in Figure 2, how to obtain the "acute cadiopulmonary process"? How to obtain the "no"?
- How to obtain the Serialized RadGraph from the Graph?
- What is the purpose of introducing a text encoder? If I understand correctly that the input is just the image, what and where is the input text?
- What is the full input of GPT-3.5 under both few-shot and zero-shot settings? Could you give an example to illustrate the full input and full output of GPT-3.5?
- How many examples are used to prompt GPT-3.5?

3. More experimental details should be provided and clarified.
- Could you give more details about the Image to Serialization task? What are the input and output of this task? What is the meaning of comparing the content of the generated serialization with the ground-truth report? Why is only the F1 score calculated? What are the results of other metrics, e.g., BELU? Therefore, it is important to clarify what are the input and output of this task.
- In Table 2, what is the baseline model? Pure Transformer or R2Gen? How many Transformer layers and the number of parameters are used in the baseline?
- Besides, while the paper reports beneficial performance, a detailed comparison with more existing image-to-report methods is missing. Including such a comparison would provide a clearer understanding of the proposed approach's advantages.
- When evaluating the AI-generated reports, are the physicians given the test set CXR images?
- The criteria used in human study is subjective. Authors are encouraged to introduce more information about three clinicians, i.e. working experience and expert title. Senior and junior clinicians may find differences when considering whether the same report is AI-generated or not.
- I strongly recommend the author to further evaluate the quality of the generated reports, instead of asking the physician to judge whether the report is AI-generated or not.



**Reproducibility:**

4: Could mostly reproduce the results, but there may be some variation because of sample variance or minor variations in their interpretation of the protocol or method.

**Reviewer Confidence:**

5: Positive that my evaluation is correct. I read the paper very carefully and I am very familiar with related work.

---

> ### Author Rebuttal · Authors · 2023-08-29
>
> We thank the reviewer for their tremendously insightful and meticulous feedback.
>
> > **(1) The work argues that existing methods, which generate reports directly from images, may lead to clinically inaccurate reports due to conflating content and style. Could you give some examples or evidence to prove it?**
>
> (Nguyen et. al 2021) can be seen as an ablation of our content extraction step, whereas style and content are kept blended together (as opposed to disentangled as in our approach). Given that Nguyen et al.’s approach is less clinically accurate, i.e., in terms of identifying clinical findings (see Table 1 and 2), this provides direct evidence supporting our hypothesis that it is valuable to model content and style separately.
> For example, here is an excerpt of a findings section of a report as compared to its  RadGraph serialization:
>
> *Report*:
> $\text{The lungs are clear. Incidentally noted are bilateral cervical ribs.}$
>
> *RadGraph Serialization*:
> $\text{lungs clear} $ \n $ \text{bilateral cervical ribs}$.
>
> Using the RadGraph Serialization can be seen as focusing the loss on the subset of tokens that represent the clinically relevant content, whereas a typical loss in prior works would treat each report token as equally important.
>
> *RadGraph Serialization as a dense subset of the report*:
> $\text{The \textcolor{green}{lungs} are \textcolor{green}{clear}. Incidentally noted are \textcolor{green}{bilateral cervical ribs}.}$
>
> In our revised draft, we will more clearly elaborate on our findings that conflating content and style leads to inferior performance.
>
>
> > **(2) Why is it more accurate to write a report that matches a physician's style? What is the advantages of usefulness of matching a physician's style?**
>
> Thank you for raising this point. We believe there are compelling reasons why explicitly modeling style could benefit radiology workflows:
>
> 1. **Disentangling style generation from content generation**: This allows us to train an initial content extraction model that is dedicated to extracting the relevant content – as opposed to being trained to jointly predict a blend of clinical content and the report style (e.g. the wording, formatting, structure etc.). Therefore, we believe that disentangling style from content allows us to focus *more* on content and clinical accuracy – not less.
>
> 2. **Intrinsic value of stylizing reports**: We believe there are a multitude of reasons why it is interesting for a model to stylize radiology reports:
>
> - *Flexibility*: Stylizing can be used to generate reports that are targeted to the readership (e.g. an orthopedic surgeon who may be interested in different aspects of a Chest X-ray than a pulmonologist). Furthermore, flexibly stylizing reports can also enable reports with less jargon that are more accessible to patients.
>
> - *Workflow*: for human-in-the-loop radiology report generation, automatically drafting a report that already closely aligns with the specific radiologist's preferred style, makes it easier / smoother for the radiologists to edit and approve the report, thereby potentially saving the radiologists valuable time, and reducing administrative burden. In our human evaluation study (Section 4.4), radiologists could not reliably distinguish between real reports and AI-generated ones in the same style - suggesting our model can closely mimic radiologist preferences. This could improve radiologist productivity and satisfaction.
>
> - *Local adaptability*: Adapting reports to the expectations of referring physicians can optimize communication of clinical findings. For instance, highlighting findings most relevant to a specialist's scope of practice (e.g., follow-up of a patch of pneumonia or correct location of
> pacing lead in the heart). In our model, this is achieved via few-shot in-context learning. Clear communication supports appropriate follow-up care.
>
> - *Consistency*: Some referring physicians may be accustomed to a particular radiologist's style of reporting. Maintaining this style in generated reports can ensure clear communication between the radiologist and the referring physician.
>
> - *Emphasis on preferred information*: Different radiologists prioritize different information (e.g. due to specialization). Tailored reports can highlight or emphasize the information that an individual radiologist finds most crucial, and can mimic the radiologist in the way this information is interpreted or highlighted.
>
> In our revised draft, we will more clearly outline the advantages of our model being able to explicitly stylize a radiology report, and to that end will extend our discussion.
>
> > **(3) How to serialize the RadGraph into a condensed summary of the clinical content of the report?**
>
> This is accounted for in Section 3.3 and Figure 2; in summary, we first partition the RadGraph into weakly connected components, corresponding to subgraphs of radiologically related and linked entities. The keywords *no* and *maybe* are prepended to entities labeled absent and uncertain in the graph, respectively. In each subgraph, a text span is formed from enumerating the entity tokens. The text spans are stratified into *Findings* and *Impressions* sections based on their referenced location in the radiology report, and concatenated into a singular, content-dense, heavily structured text.
>
> > **(4) How to ensure/define the order of entities in the generated Serialized RadGraph?**
>
> The order of entities is learned implicitly by training the image-to-serialization transformer on 213,501 reports (Line 870), analogous to how LLMs learn the proper syntax of words merely by means of next-token-prediction. The entity orderings in the training supervision (i.e. ground-truth serialization) are defined by the order of corresponding entity tokens in the report, as explained in Lines 288-290. At inference time, the order in the generated serialization is not regulated directly, and is taken as-is from the model’s prediction–to be later paraphrased by the LLM into a fully-fledged report.
>
> For future work, it will be an exciting route to investigate auxiliary loss terms to explicitly penalize the correct order of the sequence of labels (for the report generation task). Beyond the ordering of findings in available report datasets, there is no objective, canonical ordering of findings, especially when we consider graph representations thereof that possess no intrinsic ordering of the nodes and subgraph components.
>
> We will include these considerations about the Serialized RadGraph ordering in Section 3.3.
>
> > **(5) For example, in Figure 2, how to obtain the "acute cadiopulmonary process"? How to obtain the "no"?**
>
> This is achieved by the RadGraph content extractor model (Jain et. al 2021); entity extraction from the report text is heterogeneous in the sense that multiple types of entities are considered–for instance, present, ambivalent, or absent observational entities. In this case, the “no” is obtained for the model’s extraction of an absent entity. The syntagma “acute cardiopulmonary process” is obtained from the precise report tokens that are extracted as RadGraph entities.
>
> > **(6) How to obtain the Serialized RadGraph from the Graph?**
>
> The process of serializing the RadGraph into a text representation is detailed in Section 3.3 and Figure 2 of our manuscript, as well as expounded upon in our response to **Question (3)** above.
>
> > **(7) What is the purpose of introducing a text encoder? If I understand correctly that the input is just the image, what and where is the input text?**
>
> The text encoder takes in a doctor's indication, which is usually a one-sentence short description of the clinical background information of the subject of the chest x-ray study. One example indication is “x-year-old male with history of recently diagnosed pneumonia presenting with worsening fever and productive cough”. We believe that combining radiology images with the doctor's indications replicates the routine tasks of a radiologist effectively.
>
> We performed an ablation study on both the baseline model and our proposed model without the text encoder (i.e., the model input is just image). The experiments are in the Image-to-Serialization setting. Compared with the performance of models with the text encoder, we can see the benefits of doctor’s indication and the text encoder. Notably, while the clinical indication provides valuable information, we observe that leaving it away makes the relative performance gain of our approach (compared to modeling the full report directly) even more pronounced:
>
> |  Supervised Target  | Text Encoder (Clinical Indication) |   RadGraph F1  |
> |:-------------------:|:------------:|:--------------:|
> | 	Full Report 	|  	No  	| $0.128 \pm 0.004$ |
> | Serialized RadGraph |  	No  	| $0.171 \pm 0.004$ |
> | 	Full Report 	|  	Yes 	| $0.186 \pm 0.005$ |
> | Serialized RadGraph |  	Yes 	| $0.221 \pm 0.004$ |
>
> > **(8) What is the full input of GPT-3.5 under both few-shot and zero-shot settings? Could you give an example to illustrate the full input and full output of GPT-3.5?**
>
> The precise prompt structure for GPT-3.5 input is detailed in Section 3.5 and Appendix A.4. We provide an example to illustrate the full GPT-3.5 input and output in both a zero-shot and few-shot setting. Note that GPT-3.5 is a dialogue-based LLM, and prompts are structured conversationally between a system, user, and assistant.
>
> ---
> **Zero-Shot Input / Prompt**
> - System: You are a helpful assistant that generates chest x-ray reports from key words.
> - User: Generate a chest x-ray report from the following key words:  *'FINDINGS: lungs hyperinflated. no focal consolidation. no pleural effusion. no pneumothorax. Cardiac mediastinal silhouettes unremarkable. Hilar contours within normal limits. IMPRESSION: no acute cardiopulmonary process. no focal pneumonia. '*
>
> **Zero-Shot Output:** *'The chest x-ray findings reveal hyperinflated lungs without any focal consolidation, pleural effusion or pneumothorax. The cardiac mediastinal silhouettes are unremarkable and the hilar contours are within normal limits. Based on these findings, there is no acute cardiopulmonary process or focal pneumonia observed.'*
>
> ---
>
> Here, the LLM is provided with one paired input & output example of a serialization (supplied by user) and ground truth report (replied by assistant).  A $K$-shot prompt would be provided with $K$ such pairs preceding the evaluation serialization (i.e. generated from the chest X-ray at inference time).
>
> **One-Shot Input / Prompt**
> - System: You are a helpful assistant that generates chest x-ray reports from key words.
> - User: Generate a chest x-ray report from the following key words: *'FINDINGS \nleft IJ catheter upper portion SVC \nno pneumothorax \nlittle change \n'*
> - Assistant: *'In comparison with the earlier study of this date, there has been placement of a left IJ catheter that extends to the upper portion of the SVC.  No evidence of pneumothorax.  Otherwise, little change. '*
> - User: Generate a chest x-ray report from the following key words: *'FINDINGS: pacemaker leads right atrium right ventricle. position. heart size mildly enlarged. hilar mediastinal contours unremarkable. no focal consolidation. no pleural effusion. no pneumothorax. no pulmonary edema. osseous structures unremarkable. IMPRESSION: Satisfactory position pacemaker leads right atrium right ventricle. no pneumothorax. '*
>
> **One-Shot Output:** *'The chest x-ray reveals pacemaker leads in the right atrium and right ventricle, which are in a satisfactory position. The heart size is mildly enlarged and the hilar mediastinal contours are unremarkable. There is no focal consolidation, pleural effusion, or pneumothorax. Additionally, there is no pulmonary edema and the osseous structures are unremarkable. Overall, the impression is that the pacemaker leads are in a satisfactory position and there is no evidence of pneumothorax.'*
>
> ---
>
> > **(9) How many examples are used to prompt GPT-3.5?**
>
> Our study triages between 0,1,5, and 10 examples to prompt GPT-3.5, as described in Section 4.2. The results for each are provided in Tables 2 and 3 of the manuscript. We will put more emphasis on this detail in our revision by more prominently listing each number of in-context examples.
>
> > **(10) Could you give more details about the Image to Serialization task? What are the input and output of this task? What is the meaning of comparing the content of the generated serialization with the ground-truth report? Why is only the F1 score calculated? What are the results of other metrics, e.g., BELU? Therefore, it is important to clarify what are the input and output of this task.**
>
> The input is the radiology images associated with each study, and corresponding doctor’s indication (i.e., text), and the output is structured text that represents the content of the study, where we use serialized RadGraph (see Section 3.2 and 3.3 for how we obtain it) as the signal for supervision.
>
> When comparing the *content* of the generated serialization with the ground-truth report, we are looking mainly for *agreement in radiological entities and relations*—for example, the presence of pleural effusion in the lung areas–as a litmus of clinical accuracy. This is precisely what RadGraph F1 calculates; ergo it takes primacy in evaluating *content*. By contrast, NLG metrics such as BLEU and BERT score take into account non-medical and vernacular tokens, making them less ideal for gauging the accuracy of clinical content.
>
> We provide results for the other metrics below (predicted / generated serialization vs. ground-truth report on the *top row*).
>
> | Predicted           | Evaluation Target | RadCliq ($\downarrow$) | Rad F1 ($\uparrow$) | Chexbert ($\uparrow$) | BLEU ($\uparrow$) | BERT ($\uparrow$) |
> |---------------------|-------------------|-------------------|------------------|--------------------|----------------|----------------|
> | Pred Serialization  | GT Report         |   3.847 $\pm$ 0.028    |         0.221 $\pm$ 0.004           |    0.390 $\pm$ 0.008              |       0.119 $\pm$ 0.003         |          0.223 $\pm$ 0.005      |
> | Pred Report         | GT Report         | 3.530 $\pm$ 0.027 | 0.228 $\pm$ 0.004   | 0.395 $\pm$ 0.008            | 0.162 $\pm$ 0.003     | 0.332 $\pm$ 0.004     |
>
> We observe that the predicted report–the generated serialization paraphrased by an LLM–is superior in lexical metrics (BLEU, BERT), but possesses similar clinical accuracy scores (RadGraph F1, Chexbert) to the generated serialization. This is an critical observation as it prefigures the LLM’s role as rephrasing the content already present in the generated serialization, rather than creating content anew (to mitigate hallucinations).
>
>
> > **(11) In Table 2, what is the baseline model? Pure Transformer or R2Gen? How many Transformer layers and the number of parameters are used in the baseline?**
>
> The strong baseline is adapted from (Nguyen et. al 2021), with a decoder of 12 transformer blocks and an embedding dimension of 256, determined by hyperparameter sweep (detailed in Appendix A.1), and a 121-layer DenseNet vision backbone with 7.8 million parameters.
>
> The baseline possesses the identical architecture and parameters as our content generation model, as detailed in Section 3.4. The baseline is directly trained to predict the full report, while the novelty of our method is comprised of two steps: first generate clinical content in serialized RadGraph format, and then use an LLM to generate the full report based on outputs from the first step.
>
> > **(12) Besides, while the paper reports beneficial performance, a detailed comparison with more existing image-to-report methods is missing. Including such a comparison would provide a clearer understanding of the proposed approach's advantages.**
>
> The manuscript already compares the proposed approach to a strong prior method by Nguyen et al. 2021 (line 456), which was reported to outperform a large battery of over 15 prior methods; see Table 1 in (Nguyen et al. 2021). In addition, we report further results from a recently published paper (Yu et al. 2023), and compare it to our proposed approach (only the 0-shot version of our approach is directly comparable):
>
> |  Method | RadCliq ($\downarrow$) | RadGraph F1 ($\uparrow$) | CheXbert ($\uparrow$) | BLEU ($\uparrow$) | BERT Score ($\uparrow$) |
> |-----------------|--------|-----------|----------|--------------|--------|
> | R2Gen |  4.050   |  0.134 | 0.286 | 0.137 | 0.271 |
> | WCL |  3.985 |  0.143 | 0.309 | 0.144 | 0.275 |
> | CvT2DistilGPT2 | 3.903  |  0.154 | 0.335 | 0.143 | 0.280 |
> | **Ours (0-shot)** | **3.527** | **0.228** | **0.394** | **0.162** | **0.333** |
>
> All methods are evaluated on the test split of the MIMIC-CXR dataset, with identical preprocessing steps to extract findings and impressions. **We observe that our method considerably outperforms the contemporary baselines above, across all metrics including those of clinical efficacy and lexical overlap.** We will extend our paper to include these further comparisons.
>
> > **(13) When evaluating the AI-generated reports, are the physicians given the test set CXR images?**
>
> Thank you for raising this point. The “Human Style Evaluation” experiment was designed to assess whether our approach allows for the generation of radiology reports that are indistinguishably tailored to a specific radiologist’s style (e.g., $R_1$). By contrast, for assessing the clinical accuracy of the reports (i.e., whether a model extracts the correct findings from an image), we leverage established metrics such as RadGraph F1.
>
> As the reports from radiologist $R_1$ were about different patients, we believe that it is more meaningful for the human raters to focus on the reports (and their style) as there is nothing to compare by looking at the non-corresponding images. At the time of designing the experiment, we had been concerned that, if we had shown the images, the physicians would use data points from the image and not from the reports per se to make their decision.
>
> In summary, the goal of our human style evaluation was to assess the report style in isolation (without judging the report content at the same time), hence we let clinical raters review different reports by the same radiologist (plus an AI-generated one in the same style).
>
>
> > **(14) The criteria used in human study is subjective. Authors are encouraged to introduce more information about three clinicians, i.e. working experience and expert title. Senior and junior clinicians may find differences when considering whether the same report is AI-generated or not.**
>
> Thank you for raising this point. In our revised manuscript we will include further details about all the involved medical experts including years of experience and specialty.
>
> > **(15) I strongly recommend the author to further evaluate the quality of the generated reports, instead of asking the physician to judge whether the report is AI-generated or not.**
>
> This is an excellent recommendation. As illustrated in Figure 6 and Appendix A.3 of our manuscript, physicians were also asked to attribute their decisions either (1) content (e.g. missing important details or including extraneous details), (2) language (e.g. word choice, grammar, and/or writing style), or (3) structure differences with human-quality reports (e.g. different use of numbering or section headings). In A.3, we also identify and elucidate examples of correctly detected AI-generated reports (i.e. failure modes), for each of the three categories.
>
> In our revised manuscript, we will further elaborate on these results, and weigh a more comprehensive parcel of human evaluations, such as numerical ratings on the aforementioned categories et plus.
>
>
>
> **We appreciate the reviewer’s suggestions and hope that, in light of our changes to the manuscript and explanations in responses to the reviewer’s suggestions, they would consider raising their score.**
>
> *References:*
> - Feiyang et. al 2023. Evaluating progress in automatic chest X-ray radiology report generation. In Cell Patterns, 2023
> - Jain et. al 2021. Radgraph: Extracting clinical entities and relations from radiology reports. In Proceedings of the Neural Information Processing Systems Track on Datasets and Benchmarks 1, NeurIPS Datasets and Benchmarks 2021, December 2021, virtual.
> - Nguyen et. al 2021. Automated Generation of Accurate & Fluent Medical X-ray Reports. In Proceedings of the 2021 Conference on Empirical Methods in Natural Language Processing, pages 3552–3569, Online and Punta Cana, Dominican Republic. Association for Computational Linguistics.

---

### Official Review · Reviewer_aXs3 · 2023-08-04

**Soundness:** 2

**Excitement:**

2: Mediocre: This paper makes marginal contributions (vs non-contemporaneous work), so I would rather not see it in the conference.

**Paper Topic And Main Contributions:**

This study sought to utilize a novel approach in report generation tasks, generating short sentences from a RadGraph-based entity-attribute format instead of generating free-format raw text from images. The generated content was subsequently fed into the ChatGPT model, along with a few example reports, to rewrite the clinical report in a congruent style. This method of separating content and style is simple yet interesting.

However, the study lacks a compelling argument as to why considering style is "clinically important" in generating clinical reports. The more desirable approach and ultimate goal in creating clinical reports should be to focus on generating "content" from images as accurately as possible, minimizing hallucinations, and ensuring the report's factual accuracy. Therefore, the aim of research employing artificial intelligence in the medical field should not be to create slightly more natural-style reports using ChatGPT. The authors appear to have de-prioritized what is fundamentally important in the medical field, focusing on style, which makes their argument for the importance of this research less persuasive.

If the aim was to enhance the scores of widely-used metrics by considering style, the performance difference between the proposed method and the baseline was marginal or even favored the baseline in the results of the Table 2 experiment. Specifically, it was not surprising to see that the scores for RadCliQ and RadGraph were better, given that the method used RadGraph in the training phase. Also, a slight difference was evident in the chexbert result. The absence of comparison with other studies further complicates the understanding of this study's significance.

The authors attempted to address the problem of generating clinical reports from a new perspective. However, the study lacks an explanation of the clinical significance and implications of this research, and the results are not particularly surprising.

**Reasons To Accept:**

The authors attempted to address the issue of clinical report generation from a new perspective in their methodology, and they achieved results indicating that ChatGPT can reference style well enough to generate reports that are indistinguishable from those written by humans.

**Reasons To Reject:**

1. This study doesn't persuasively explain "why style should be considered in the medical domain" as proposed by this study, and it doesn't sufficiently explore the clinical importance of such a consideration.

2. Experimental results lack comparisons with other studies. (It's necessary to verify statistically significant results by conducting repeated experiments on the same data set.)

3. While this study incorporates a few ideas and uses RadGraph and ChatGPT, it does not offer considerable novelty in its methods or contribute significantly to the field.


**Reproducibility:**

5: Could easily reproduce the results.

**Reviewer Confidence:**

4: Quite sure. I tried to check the important points carefully. It's unlikely, though conceivable, that I missed something that should affect my ratings.

---

> ### Author Rebuttal · Authors · 2023-08-29
>
> We thank the reviewer for the valuable feedback that helped us substantially improve our manuscript.
>
> > **However, the study lacks a compelling argument as to why considering style is "clinically important" in generating clinical reports. The more desirable approach and ultimate goal in creating clinical reports should be to focus on generating "content" from images as accurately as possible, minimizing hallucinations, and ensuring the report's factual accuracy.**
>
> Thank you for raising this point. However, we believe there are compelling reasons why explicitly modeling style could benefit radiology workflows:
>
> 1. **Disentangling style generation from content generation**: This allows us to train an initial content extraction model that is dedicated to extracting the relevant content – as opposed to being trained to jointly predict a blend of clinical content and the report style (e.g. the wording, formatting, structure etc.). Therefore, we believe that disentangling style from content allows us to focus *more* on content and clinical accuracy – not less.
>
> 2. **Intrinsic value of stylizing reports**: We believe there are a multitude of reasons why it is interesting for a model to stylize radiology reports:
>
> - *Flexibility*: Stylizing can be used to generate reports that are targeted to the readership (e.g. an orthopedic surgeon who may be interested in different aspects of a Chest X-ray than a pulmonologist). Furthermore, flexibly stylizing reports can also enable reports with less jargon that are more accessible to patients.
>
> - *Workflow*: for human-in-the-loop radiology report generation, automatically drafting a report that already closely aligns with the specific radiologist's preferred style, makes it easier / smoother for the radiologists to edit and approve the report, thereby potentially saving the radiologists valuable time, and reducing administrative burden. In our human evaluation study (Section 4.4), radiologists could not reliably distinguish between real reports and AI-generated ones in the same style - suggesting our model can closely mimic radiologist preferences. This could improve radiologist productivity and satisfaction.
>
> - *Local adaptability*: Adapting reports to the expectations of referring physicians can optimize communication of clinical findings. For instance, highlighting findings most relevant to a specialist's scope of practice (e.g., follow-up of a patch of pneumonia or correct location of
> pacing lead in the heart). In our model, this is achieved via few-shot in-context learning. Clear communication supports appropriate follow-up care.
>
> - *Consistency*: Some referring physicians may be accustomed to a particular radiologist's style of reporting. Maintaining this style in generated reports can ensure clear communication between the radiologist and the referring physician.
>
> - *Emphasis on preferred information*: Different radiologists prioritize different information (e.g. due to specialization). Tailored reports can highlight or emphasize the information that an individual radiologist finds most crucial, and can mimic the radiologist in the way this information is interpreted or highlighted.
>
> In our revised draft, we will more clearly outline the advantages of our model being able to explicitly stylize a radiology report, and to that end will extend our discussion.
>
> > **The authors appear to have de-prioritized what is fundamentally important in the medical field, focusing on style, which makes their argument for the importance of this research less persuasive.**
>
> Thank you for raising this point. Our narrative is not to simply “focus on style”. Instead, our line of reasoning is the following: We disentangle content from style. This allows for dedicated optimization of clinical content (content extraction), and style generation (via LLM few-shot adaptations). We find that training our model on dense clinical content allows us to improve in clinical accuracy (as opposed to “de-priotizing it”). Moreover, we also evaluate content and style in different settings to scrutinize individual steps our pipeline (e.g., image-to-serialization, serialization-to-report, end-to-end report generation).
> We will update our introduction to make it more clear to the reader that our paper aims at disentangling content from style in order to optimize *both aspects* separately, which we find leads to improved performance.
>
> > **If the aim was to enhance the scores of widely-used metrics by considering style, the performance difference between the proposed method and the baseline was marginal or even favored the baseline in the results of the Table 2 experiment. Specifically, it was not surprising to see that the scores for RadCliQ and RadGraph were better, given that the method used RadGraph in the training phase. Also, a slight difference was evident in the chexbert result.**
>
> Thank you for raising this point; we would like to clarify. First, the RadGraph (Jain et. al 2021) is extracted directly from the report tokens (that is, graph nodes constitute a subset of tokens in the report, and graph edges are based solely on syntactic and morphological relations in the report text), which are equally visible to the baseline model during training—its supervision target being the entire, fully-fledged report. Thus, our method does not have access to additional medical information that would privilege it in measures of clinical accuracy. Rather, the information is organized in a more structured and accessible way.
>
> Table 2 shows that our method outperforms the (Nguyen et al. 2021) baseline across all clinical accuracy metrics: RadCliQ, an aggregate metric that by design is closely aligned with the preference of radiologists (Yu et. al 2023), RadGraph F1, and CheXbert score. The remaining, more classical NLG metrics are substantially less reliable in the clinical setting (for example, changing a single word may change the meaning and implications of a medical paragraph), which is why non-superiority on these additional measures is not clinically relevant. Indeed, (Yu et. al 2023) finds that metrics such as BLEU poorly approximate radiologists’s evaluation of reports, in comparison to clinical accuracy metrics.
>
> Furthermore, in our response to the next point below, we show further comparisons to prior works with convincing performance gains. We will update our result section to better explain the relevance of the clinical metrics (in contrast to metrics like BLEU), and the further comparisons to prior work.
>
> > **Experimental results lack comparisons with other studies. (It's necessary to verify statistically significant results by conducting repeated experiments on the same data set.)**
>
> We concur that such comparisons are necessary; our paper does include both quantitative and qualitative comparisons to other studies:
> - In our quantitative evaluations (Table 2), we compare our approach to a strong baseline model adapted from (Nguyen et al. 2021). We show that our approach outperforms this prior work in clinical accuracy metrics like RadGraph F1 and CheXbert similarity while achieving competitive performance in natural language metrics (NLG) like BLEU. Furthermore, in accordance with (Yu et al. 2023), NLG metrics such as BLEU have been found to poorly correlate with radiologists’ evaluations of reports compared with clinical accuracy metrics.
> - We compare our approach to further prior works by leveraging directly comparable results from a recently published paper (Yu et al. 2023) (only the 0-shot version of our approach is directly comparable).
>
> |  Method | RadCliq ($\downarrow$) | RadGraph F1 ($\uparrow$) | CheXbert ($\uparrow$) | BLEU ($\uparrow$) | BERT Score ($\uparrow$) |
> |-----------------|--------|-----------|----------|--------------|--------|
> | R2Gen |  4.050   |  0.134 | 0.286 | 0.137 | 0.271 |
> | WCL |  3.985 |  0.143 | 0.309 | 0.144 | 0.275 |
> | CvT2DistilGPT2 | 3.903  |  0.154 | 0.335 | 0.143 | 0.280 |
> | **Ours (0-shot)** | **3.527** | **0.228** | **0.394** | **0.162** | **0.333** |
>
> All methods are evaluated on the test split of the MIMIC-CXR dataset, with identical preprocessing steps to extract findings and impressions for joint generation. **We observe that our method considerably outperforms the contemporary baselines above, across all metrics including those of clinical efficacy and lexical overlap.** We will extend our paper to include these salient further comparisons.
> - In the human evaluation study (Section 4.4), we have radiologists write reports in their own style, and then compare these to AI-generated reports that are adapted to the radiologist’s style (via in-context learning over serialization-report pairs). Showing that human experts cannot reliably distinguish between real radiologist reports and AI-generated ones (that mimic their style) demonstrates the quality of our approach to tailor radiology report generation to individual radiologists.
>
> To summarize, we believe the current manuscript provides adequate qualitative and quantitative comparisons to prior studies. We find our method outperforms a suite of recent works (elucidated in the table above), as well as the strong (Nguyen et. al 2021) baseline in terms of clinical accuracy, which–we agree–has evaluation primacy.
>
> > **While this study incorporates a few ideas and uses RadGraph and ChatGPT, it does not offer considerable novelty in its methods or contribute significantly to the field.**
>
> The core novelty of our work lies in the strategy of disentangling content and style through a two-step generation process. Additionally, we make several other novel contributions around supervision, evaluation, and the applicability of large language models. We believe these collectively constitute meaningful advances to the field.
>
> In particular:
> 1. We propose a new two-step strategy for radiology report generation that disentangles content generation from style injection. Treating these as separate steps is novel.
> 2. For content generation, we supervise on serialized RadGraphs rather than full reports. Using dense structured representations for supervision enables a focus on clinical accuracy.
> 3. We demonstrate that serializing RadGraphs is an effective bridge between vision and language modalities for content extraction.
> 4. Our approach allows flexible conditioning of style based on only a few examples, by leveraging large language models. Style adaptation is not well explored in previous work.
> 5. To our knowledge, we are the first to evaluate style transfer for radiology reports by having clinical experts attempt to detect AI-generated reports. This provides a rigorous human evaluation of style quality.
>
> **In light of these clarifications and the additional context we have provided around the novelty and value of our approach, we kindly request that you reconsider the scores assigned to our paper. We believe the revisions detailed above address the limitations you noted, and help establish the unique contributions and clinical relevance of our work. Specifically, we feel the novelty of our two-step disentangled approach to separate content extraction from style generation, the use of serialized RadGraphs for supervision, and the rigorous human evaluation study all represent meaningful advances.**
>
> **We hope you agree that the additional details provided in this response help to position our work as making valuable contributions to the field of natural language processing for radiology report generation. We would be grateful if you could raise the scores accordingly to better reflect the merits of our approach after accounting for these clarifications. We appreciate you taking the time to re-evaluate our work, and look forward to receiving your feedback.**
>
>
> *References:*
>
> - Feiyang et. al 2023. Evaluating progress in automatic chest X-ray radiology report generation. In Cell Patterns, 2023
> - Jain et. al 2021. Radgraph: Extracting clinical entities and relations from radiology reports. In Proceedings of the Neural Information Processing Systems Track on Datasets and Benchmarks 1, NeurIPS Datasets and Benchmarks 2021, December 2021, virtual.
> - Nguyen et. al 2021. Automated Generation of Accurate & Fluent Medical X-ray Reports. In Proceedings of the 2021 Conference on Empirical Methods in Natural Language Processing, pages 3552–3569, Online and Punta Cana, Dominican Republic. Association for Computational Linguistics.

---

### Official Review · Reviewer_5JVq · 2023-08-10

**Soundness:** 3

**Excitement:**

4: Strong: This paper deepens the understanding of some phenomenon or lowers the barriers to an existing research direction.

**Paper Topic And Main Contributions:**


- The authors introduce a novel method for generating reports emphasizing style awareness.
- The proposed model follows a two-step approach to separate report content and style.
- In the initial phase, the model is trained to produce serialized data that captures the clinical content of the report.
- Subsequently, a Language Model (LLM) is employed to convert the predicted serialized data into a coherent report.
- Importantly, this transformation process ensures that the resulting report aligns seamlessly with the distinct style preferences of a designated radiologist, achieved through in-context learning techniques.

**Reasons To Accept:**


- The authors propose a straightforward yet effective approach that involves converting raw reports into structured data. This process aids in the removal of non-semantic content.
- They also introduce a style-aware report generation technique and carry out a human evaluation to assess its effectiveness.
- Easy to follow

**Reasons To Reject:**

- Is there no loss of information from the Serialization to Report process? What are your thoughts on the reason behind the slight difference in performance, with a RadGraphF1 of 0.221 ± 0.004 for Table1 and 0.228 ± 0.004 for Table2?

- In the Image Encoder section (Section 3.4), it is mentioned that if there are multiple images in one study, they are aggregated. It would be beneficial to evaluate the test performance based on the number of images present in each study, showcasing the performance for cases where a study contains one image, two images, and so on.

- In the Content Generation Model section (Section 3.4), it is mentioned that the Text Encoder receives the clinical document as input, but the specific details are not provided.

- It would be valuable to experiment with both the Baseline and the Proposed Model without the Text Encoder and share the results.
This paper does not propose a new architectural design but shows the effectiveness of serialized data.
Given this focus, I think that showcasing results across various scenarios would be valuable to illustrate the impact more comprehensively.

- In the 4.2 Serialization to Report experiment, the zero-shot performance is observed to be 0.722 RadGraphF1.
Despite using ground-truth serialization, the performance is not nearly perfect.
Is it plausible to attribute this discrepancy to the fact that the performance of the content extractor (Report -> RadGraph preprocessing) might not be flawless?

**Reproducibility:**

4: Could mostly reproduce the results, but there may be some variation because of sample variance or minor variations in their interpretation of the protocol or method.

**Reviewer Confidence:**

4: Quite sure. I tried to check the important points carefully. It's unlikely, though conceivable, that I missed something that should affect my ratings.

---

> ### Author Rebuttal · Authors · 2023-08-29
>
> We thank the reviewer for their exemplary and insightful comments.
> > **Is there no loss of information from the Serialization to Report process? What are your thoughts on the reason behind the slight difference in performance, with a RadGraphF1 of 0.221 ± 0.004 for Table1 and 0.228 ± 0.004 for Table2?**
>
> Table 1 reflects the performance of the predicted serialization against the ground truth report; Table 2  reflects the performance of the predicted report—the predicted serialization paraphrased by an LLM to improve lexical coherence. The slight difference of +0.007 thus results from attuned wording and syntax. This can improve readability to the machine parser involved in calculating RadGraph F1, based on parsed radiological entities. We also provide an expanded table with the other metrics.
>
> | Predicted           | Evaluation Target | RadCliq ($\downarrow$) | Rad F1 ($\uparrow$) | Chexbert ($\uparrow$) | BLEU ($\uparrow$) | BERT ($\uparrow$) |
> |---------------------|-------------------|-------------------|------------------|--------------------|----------------|----------------|
> | Pred Serialization  | GT Report         |   3.847 $\pm$ 0.028    |         0.221 $\pm$ 0.004           |    0.390 $\pm$ 0.008              |       0.119 $\pm$ 0.003         |          0.223 $\pm$ 0.005      |
> | Pred Report         | GT Report         | 3.530 $\pm$ 0.027  | 0.228 $\pm$ 0.004    | 0.395 $\pm$ 0.008            | 0.162 $\pm$ 0.003     | 0.332 $\pm$ 0.004     |
>
> This supports the claim that there is minimal–if any–clinical accuracy loss (Rad F1, Chexbert) from serialization to report, a step that functions primarily to reword the serialized content into an acceptable prosaic quality, rather than generate content anew (to mitigate hallucinations). Furthermore, the table shows that comparing the fully-fletched predicted report with the ground-truth report results in high NLG metric scores (BLEU, BERT) likely attributed to the increased fluency of the predicted report (as compared to the serialization). This expected behaviour therefore also affects the aggregate metric RadCliQ which is a linear combination of the other metrics. We see that serialized text is–by itself–an inadequate substitute for a fully-fledged report, despite capturing relevant clinical information.
>
>
> > **In the Image Encoder section (Section 3.4), it is mentioned that if there are multiple images in one study, they are aggregated. It would be beneficial to evaluate the test performance based on the number of images present in each study, showcasing the performance for cases where a study contains one image, two images, and so on.**
>
> We concur with the benefits of this evaluation, and have added the table below for reports generated under the zero-shot setting (i.e. no in-context learning examples).
>
> | # of Images | n (# of studies) | RadCliq ($\downarrow$) | Rad F1 ($\uparrow$) | Chexbert ($\uparrow$) | BLEU ($\uparrow$) | BERT ($\uparrow$) |
> |-------------|------------------|-------------------|------------------|--------------------|----------------|----------------|
> | 1           | 1439             | 3.586          | 0.218         | 0.387           | 0.155       | 0.322       |
> | 2           | 1065             | 3.471          | 0.246         | 0.401         | 0.172       | 0.343       |
> | 3           | 265              | 3.463          | 0.232         | 0.411           | 0.166       | 0.343       |
> | 4           | 29               | 3.533          | 0.217         | 0.378           | 0.180       | 0.352       |
> | 5           | 1                | 2.819          | 0.368         | 0.620           | 0.295       | 0.386       |
>
>
> We find that multiple images (>1) correspond generally with enhanced evaluation metrics. However, this finding is complicated by the unequal distribution of image quantities—glaringly with the singular 5-image study—as demarcated by the external MIMIC-CXR dataset.
>
> > **In the Content Generation Model section (Section 3.4), it is mentioned that the Text Encoder receives the clinical document as input, but the specific details are not provided.**
>
> We thank the reviewer for catching this and will clarify the following in the paper. The text encoder takes in a doctor's indication, which is usually a one-sentence short description of the clinical background information of the subject of the chest x-ray study. One example indication is “x-year-old male with history of recently diagnosed pneumonia presenting with worsening fever and productive cough”. We believe that combining radiology images with the doctor's indications replicates the routine tasks of a radiologist effectively.
> The text embedding generated by the text encoder (detailed on line 334-351) will be concatenated with the visual embedding through summation and layer normalization, and this fused embedding will be used as the input for the report decoder (detailed on line 352-370).
>
> > **It would be valuable to experiment with both the Baseline and the Proposed Model without the Text Encoder and share the results. This paper does not propose a new architectural design but shows the effectiveness of serialized data. Given this focus, I think that showcasing results across various scenarios would be valuable to illustrate the impact more comprehensively.**
>
> We appreciate the reviewer’s suggestion and ran the Image-to-Serialization experiments with both the baseline model and the proposed model without the text encoder (i.e. the model only takes images as input) and present the results below. This table also demonstrates the benefits of using doctor’s indications and the text encoder.  We can also see that our proposed model, using the serialized RadGraph as the supervision target, still achieves a better result compared to the baseline model in the absence of the text encoder. Notably, in the absence of the clinical indication, the benefit of our approach compared to training on the full report is even more pronounced (33.6% gain in RadGraph F1 without indication versus 18.8% gain when using the indication).
>
> |  Supervised Target  | Text Encoder (Clinical Indication) |   RadGraph F1  |
> |:-------------------:|:------------:|:--------------:|
> |     Full Report     | 	 No 	 | 0.128 $\pm$ 0.004 |
> | Serialized RadGraph | 	 No 	 | 0.171 $\pm$ 0.004 |
> |     Full Report     | 	 Yes     | 0.186 $\pm$ 0.005 |
> | Serialized RadGraph | 	 Yes     | 0.221 $\pm$ 0.004 |
>
> > **In the 4.2 Serialization to Report experiment, the zero-shot performance is observed to be 0.722 RadGraphF1. Despite using ground-truth serialization, the performance is not nearly perfect. Is it plausible to attribute this discrepancy to the fact that the performance of the content extractor (Report -> RadGraph preprocessing) might not be flawless?**
>
> This is a salient point of inquiry. As mentioned in our limitations section, the RadGraph extraction from the report is based on a pre-trained DYGIE++ model (Jain et. al 2021), which is not flawless despite generally strong performance in relation and entity extraction. Beyond limitations in the content extractor, it is conceivable that the LLM rephrasing could introduce some noise. As an ablative test, we compare the ground-truth serialization directly with the ground-truth report, without the LLM to paraphrase the serialization into a zero-shot report.
>
> | Prediction      | Target   | RadGraph F1 |
> |-----------------|----------|-------------|
> | GT Serialization| GT Report|              0.632 $\pm$ 0.005 |
> | Zero-Shot Report| GT Report|             0.722 $\pm$ 0.006 |
>
> With comparable–even better–performance in RadGraph F1 with zero-shot reports, we can largely exonerate LLM hallucination as a cause of zero-shot losses. Instead, the LLM’s enhancements in report readability and wording were successful at improving clinical coherence. The results suggest the RadGraph content extractor is the rate-limiting step here; an enhanced extractor would be portent for higher accuracy.
>
>
> **We appreciate the reviewer’s suggestions and hope that, in light of our changes to the manuscript and explanations in responses to the reviewer’s suggestions, they would consider raising their score.**
>
> *References*:
>
> Saahil Jain, et al. 2021. Radgraph: Extracting clinical entities and relations from radiology reports. In Proceedings of the Neural Information Processing Systems Track on Datasets and Benchmarks 1, NeurIPS Datasets and Benchmarks 2021, December 2021, virtual.

---

### Meta-Review · Area_Chair_UAFt · 2023-09-18

**Recommendation:** 2

**Metareview:**

This paper proposes a two-step approach to automatically generating radiology reports, where the first stage focuses on extracting content from the image and the second stage verbalizing the content to a specific style. Although the reviewers agreed that the proposed approach is straightforward and convincing, a strong concern remained regarding the fact that the experiment results did not substantiate the core claim of disentangling the content and style of radiology reports, and the potential risk of missing important content while verbalizing it to different styles.

---

### Decision · Program_Chairs · 2023-10-07

**Decision:**

Accept-Findings

**Comment:**

This paper proposes a two-step approach to automatically generating radiology reports, where the first stage focuses on extracting content from the image and the second stage verbalizing the content to a specific style. Although the reviewers agreed that the proposed approach is straightforward and convincing, a strong concern remained regarding the fact that the experiment results did not substantiate the core claim of disentangling the content and style of radiology reports, and the potential risk of missing important content while verbalizing it to different styles.